# Activation-Guided Regularization: Improving Deep Classifiers using Feature-Space Regularization with Dynamic Prototypes

## Abstract

The softmax cross-entropy loss, which is the de facto standard for training deep classifiers, does not explicitly guide the formation of a well-structured internal feature space. This can limit model generalization and robustness. In this paper, we explore how the deep learning model's internal neuron activation patterns can be leveraged to create a powerful regularization signal. We introduce Activation Guided Regularization (AGR), a novel training objective that directly addresses this. AGR enhances standard training by introducing a secondary objective that encourages a sample's feature embedding to be similar to a dynamically generated prototype. These prototypes, which represent the mean neuron activation pattern for each class using the model's own high-confidence predictions, are generated dynamically in an efficient, self-regularizing feedback loop that requires no changes to the model architecture. We conduct extensive experiments across diverse computer vision benchmarks, including standard object recognition, fine-grained classification, and medical imaging tasks. Our results demonstrate that AGR consistently and significantly improves classification accuracy over strongly-trained baselines across a variety of architectures, from simple CNNs to large transformer-based models. Furthermore, we provide extensive analysis showing that these performance gains are a direct result of a more structured learned feature space, characterized by quantitatively improved intra-class compactness, inter-class separability, and qualitatively clearer cluster separation in t-SNE and UMAP visualizations. Finally, we show that these superior representations, learned by attending to neuron activation patterns, lead to enhanced model robustness against data corruptions and improved feature transferability in few-shot learning scenarios.

## 1 Introduction

The remarkable success of deep neural networks is largely attributed to their ability to learn rich, hierarchical feature representations directly from data (LeCun et al., 2015). However, the standard training paradigm, which relies on minimizing a cross-entropy loss, provides no explicit signal to encourage a well-structured internal feature space. While this approach is effective at reducing prediction error on the training set, it can result in representations that are not optimally organized, leading to sub-optimal generalization and a lack of robustness to domain shifts and data corruptions (Arjovsky et al., 2019). A central goal of representation learning is to learn a feature space in which same-class representations are compact and different-class representations are well separated (Fisher, 1936; Wen et al., 2016; Liu et al., 2017; Deng et al., 2019).

A significant body of work in metric and contrastive learning has sought to achieve this by directly shaping the feature space, often through losses that require careful and computationally expensive sampling of pairs or triplets (Schroff et al., 2015; He et al., 2020; Chen et al., 2020). While powerful, the complexity and sensitivity of these sampling strategies pose a barrier to their broad application. A common way to sculpt embedding structure uses example–example objectives (contrastive/triplet/N-pair), which depend on curated positive/negative selection and often incur tuning and computational overhead (Hadsell et al., 2006; Schroff et al., 2015; Sohn, 2016; Hermans et al., 2017; Wu et al., 2017). This raises a critical question: can we regularize the embedding space us-

ing *class-conditioned centroids* rather than example–example comparisons, thereby avoiding mining heuristics and large-batch requirements?

In this paper, we answer the question in the affirmative and introduce a simple yet powerful training paradigm that we call **Activation-Guided Regularization (AGR)**. AGR enhances standard classification training by leveraging the model's own evolving understanding of the data. The core of our method is a feedback loop where we dynamically generate class prototypes from the neuron activation patterns of high-confidence predictions. We then introduce a secondary loss objective that encourages the features of a given sample to be similar to its corresponding class prototype. This acts as a powerful regularizer, guiding the model to learn a more coherent and geometrically sound feature space. Our primary contribution is a novel and efficient loss formulation that directly encourages intra-class compactness by comparing samples to *adaptive prototypes* that co-evolve with the model's feature space, without requiring complex pair or triplet mining. Here, a *prototype* for class $c$ is a confidence-weighted centroid in the penultimate-layer embedding space, implemented as an exponential moving average of features for that class. We provide extensive empirical evidence demonstrating that AGR consistently improves generalization and robustness. Our method yields significant accuracy gains across a wide range of benchmarks and architectures, and quantitative analyses show that these gains stem from a more structured feature space—exhibiting lower intra-class dispersion, higher inter-class separation, greater robustness to corruptions, and stronger few-shot transfer.

## 2 Related Work

Our work lies at the intersection of representation learning, loss design, and analysis of internal network representations.

### 2.1 Loss Functions for Deep Image Classification

The common objective function for training deep classifiers is softmax cross-entropy. While highly effective, it optimizes class likelihood rather than explicitly structuring the embedding space (Ye et al., 2022). A large body of work modifies cross-entropy to improve training dynamics—e.g., Label Smoothing (Szegedy et al., 2016), Focal Loss for class imbalance (Lin et al., 2017), and losses robust to label noise (Cheng & Zhang, 2023). A complementary thread explicitly shapes feature geometry via *margin* or *center* terms, such as Center Loss (Wen et al., 2016) and angular-margin variants (SphereFace, CosFace, ArcFace) (Liu et al., 2017; Deng et al., 2019), which enlarge inter-class separation in angular space while tightening intra-class spread. AGR differs in mechanism: instead of modifying the classifier margin or introducing learned centers, it uses *online, confidence-weighted class centroids* (EMA of penultimate-layer embeddings) as a *regularizing target* during training, leaving the classifier head unchanged.

### 2.2 Metric and Representation Learning

Metric learning directly optimizes distances so that similar examples are close and dissimilar ones are far. Classical objectives include contrastive and triplet losses (Hadsell et al., 2006; Schroff et al., 2015), with extensive work on mining and sampling (Hermans et al., 2017; Wu et al., 2017) and $N$-pair formulations (Sohn, 2016). Proxy-based methods learn per-class vectors to avoid pair mining (e.g., Proxy-NCA, Proxy-Anchor). Few-shot *Prototypical Networks* (Snell et al., 2017) use class means and nearest-prototype classification in episodic training. AGR is closest in spirit to centroid ideas but (i) uses *non-parametric class centroids derived from high-confidence predictions* rather than learned proxies, and (ii) employs them *as a regularizer alongside cross-entropy* rather than as the classifier or sole supervision. Self-supervised contrastive learning (SimCLR, MoCo) and modern variants (DINOv2) learn strong representations via contrasting augmented views (Chen et al., 2020; He et al., 2020; Oquab et al., 2023); supervised contrastive learning extends this to label-aware positives (Khosla et al., 2020). These methods typically rely on heavy augmentations, large batches/queues, and example–example objectives; AGR instead supplies a supervised, augmentation-agnostic, class-conditional regularization signal without pairwise sampling.

## 2.3 Analyses of Internal Representations and Activations

Internal representations have been probed for interpretability and comparison: Network Dissection identifies concept-selective units (Bau et al., 2017), TCAV quantifies concept influence (Kim et al., 2018), and SVCCA/CKA compare layerwise geometry across models (Raghu et al., 2017; Kornblith et al., 2019). Attribution methods such as occlusion, Grad-CAM, and Integrated Gradients assess input–output sensitivity; recent studies link activation statistics to capacity/data (Pope et al., 2023) and surface subgroups in medical imaging (Wong et al., 2024). AGR aligns with the view that *internal activations carry class-conditional structure*, but uses activation-derived centroids *during training* to directly regularize the embedding space rather than solely for post-hoc analysis.

Compared to margin losses, proxy or prototype-based metric learning, and contrastive objectives, AGR provides a lightweight, architecture-agnostic regularizer that (i) uses non-parametric class centroids computed from penultimate-layer embeddings, (ii) avoids example–example objectives and mining heuristics, and (iii) keeps the classifier head unchanged while improving intra-class compactness and inter-class separation.

## 3 Methodology

Deep neural networks are typically trained by minimizing the categorical cross-entropy loss, computed exclusively on the output logits. While effective for classification, this paradigm provides no explicit signal to structure the intermediate feature space. Consequently, representations may lack intra-class compactness or inter-class separation, which limits generalization and robustness. To address this, we introduce Activation-Guided Regularization (AGR), a loss function that augments standard training with an additional term derived from the model's internal activation patterns. AGR dynamically enforces compact, discriminative representations by leveraging class prototypes computed during training.

### 3.1 Influence of AGR on Training

Let $\phi_\theta(\mathbf{x}) \in \mathbb{R}^d$ denote the feature representation of input $\mathbf{x}$ under network parameters $\theta$. The goal of a classifier is to learn a mapping such that samples of the same class cluster tightly, while different classes remain well-separated. This can be formalized using the within-class and between-class scatter matrices:

$$S_W(\theta) = \mathbb{E}_{(\mathbf{x},y)}\big[(\phi_\theta(\mathbf{x}) - \boldsymbol{\mu}_y)(\phi_\theta(\mathbf{x}) - \boldsymbol{\mu}_y)^\top\big], \tag{1}$$

$$S_B(\theta) = \sum_{c=1}^{C} \pi_c \, (\boldsymbol{\mu}_c - \boldsymbol{\mu})(\boldsymbol{\mu}_c - \boldsymbol{\mu})^\top, \tag{2}$$

where $\boldsymbol{\mu}_c = \mathbb{E}_{\mathbf{x} \sim P_c}[\phi_\theta(\mathbf{x})]$ is the class-$c$ mean in embedding space, $\boldsymbol{\mu} = \mathbb{E}_{(\mathbf{x},y)}[\phi_\theta(\mathbf{x})]$ is the global mean, and $\pi_c$ is the (scalar) prior of class $c$. Here $(\mathbf{x}, y) \sim \mathcal{D}$ denotes a sample from the data distribution with $y \in \{1, \ldots, C\}$. We write $\boldsymbol{\mu}_y$ for the class mean associated with label $y$ (i.e., $\boldsymbol{\mu}_y = \boldsymbol{\mu}_c$ when $y = c$). In $S_B$, $(\boldsymbol{\mu}_c - \boldsymbol{\mu})(\boldsymbol{\mu}_c - \boldsymbol{\mu})^\top$ is an outer product; the order is standard and $\pi_c$ is a scalar multiplier. The Fisher Discriminant Ratio is:

$$J(\theta) \;=\; \frac{\mathrm{tr}\big(S_B(\theta)\big)}{\mathrm{tr}\big(S_W(\theta)\big)}. \tag{3}$$

Standard cross-entropy loss encourages large $\mathrm{tr}(S_B)$ by pushing class logits apart, but it does not explicitly reduce $\mathrm{tr}(S_W)$, where $\mathrm{tr}(\cdot)$ denotes the matrix trace. AGR indirectly reduces intra-class variance by pulling features toward their prototypes, thereby encouraging tighter clusters. This effect can be understood as indirectly maximizing $J(\theta)$ (Eq. 3); we make it explicit via the consistency term in Eq. 7.

### 3.2 Activation-Guided Regularization

AGR modifies the standard training pipeline by introducing a prototype-driven similarity distribution, which is blended with the model's logits to form a regularized prediction.

---

**Algorithm 1** Activation-Guided Regularization (AGR) Training

---

**Require:** Model $f_\theta$, dataset $\mathcal{D}$, warm-up epochs $E_{\text{warmup}}$, total epochs $E_{\text{total}}$, momentum $\beta$, temperature $\tau$, weight $\lambda$, blend $\alpha$
  **for** epoch $e = 1$ to $E_{\text{total}}$ **do**
    **if** $e < E_{\text{warmup}}$ **then**
      Train $f_\theta$ on $\mathcal{D}$ using cross-entropy only
    **else**
      **for** each minibatch $(\mathbf{x}_b, \mathbf{y}_b)$ of size $B$ **do**
        Extract embeddings $\mathbf{\Phi}_b \leftarrow \phi_\theta(\mathbf{x}_b)$
        Update prototypes $\{\boldsymbol{\mu}_c\}$ using $\mathbf{\Phi}_b$ with confidence-weighted updates (keep $\boldsymbol{\mu}_c$ if no reliable samples)
        Compute network probs $\hat{\mathbf{Y}}_b$ and prototype probs $\hat{\mathbf{Y}}_b^{\text{sim}}$
        Blend $\hat{\mathbf{Y}}_b^{\text{prop}} \leftarrow \alpha\,\hat{\mathbf{Y}}_b + (1-\alpha)\,\hat{\mathbf{Y}}_b^{\text{sim}}$
        Compute $\mathcal{L}_{\text{AGR}} \leftarrow \text{CE}\big(\hat{\mathbf{Y}}_b^{\text{prop}}, \mathbf{y}_b\big) \; + \; \dfrac{\lambda}{B}\sum_{i=1}^{B}\big\|\mathbf{\Phi}_{b,i} - \boldsymbol{\mu}_{y_i}\big\|^2$
        Update $\theta$ using gradients of $\mathcal{L}_{\text{AGR}}$
      **end for**
      Decay $\alpha$ adaptively (e.g., exponential or cosine schedule)
    **end if**
  **end for**

---

**Dynamic Prototype Update.** For each class $c$, we maintain a prototype $\boldsymbol{\mu}_c \in \mathbb{R}^d$. Given a minibatch $\{(\mathbf{x}_i, y_i)\}_{i=1}^{B}$ with predictions $\hat{y}_i = f_\theta(\mathbf{x}_i)$ and $p_{ic} = [\hat{\mathbf{y}}_i]_c$, we form a *reliable* set using top-1 agreement ($y_i = c$ and $\arg\max_k[\hat{\mathbf{y}}_i]_k = c$). Let $\tilde{\phi}_\theta(\mathbf{x}_i) = \phi_\theta(\mathbf{x}_i)/\|\phi_\theta(\mathbf{x}_i)\|$ be L2-normalized features. If at least $m$ such samples exist for class $c$ in the batch (we use $m = 50$), we compute a confidence-weighted centroid and momentum-update the prototype; otherwise we update a fallback estimate and keep $\boldsymbol{\mu}_c$ unchanged. A class-balancing pass ensures that classes with few updates inherit their fallback estimate so that all classes maintain usable prototypes.

**Feature Similarity Distribution:** For an input $\mathbf{x}_i$, we compute cosine similarity between its normalized feature vector and the prototypes:

$$s_{ic} \;=\; \frac{\phi_\theta(\mathbf{x}_i) \cdot \boldsymbol{\mu}_c}{\|\phi_\theta(\mathbf{x}_i)\|\,\|\boldsymbol{\mu}_c\|}. \tag{4}$$

These scores are transformed into a probability distribution via a softmax with temperature $\tau$:

$$\hat{y}_{ic}^{\text{sim}} \;=\; \frac{\exp(s_{ic}/\tau)}{\sum_{c'=1}^{C}\exp(s_{ic'}/\tau)}. \tag{5}$$

We further apply inverse-frequency reweighting of class logits (based on running class update counts) to mitigate bias toward frequent classes before the softmax is taken.

**Adaptive Blending:** The final prediction is a convex combination of the network's softmax prediction $\hat{\mathbf{y}}_i$ and the prototype similarity distribution $\hat{\mathbf{y}}_i^{\text{sim}}$:

$$\hat{\mathbf{y}}_i^{\text{prop}} \;=\; \alpha\,\hat{\mathbf{y}}_i + (1-\alpha)\,\hat{\mathbf{y}}_i^{\text{sim}}, \tag{6}$$

where $\alpha \in [0,1]$ is decayed over epochs, e.g., $\alpha_e = \alpha_0 \cdot \gamma^e$ or via a cosine schedule, so reliance on prototype guidance increases as prototypes stabilize, following curriculum-style weighting strategies (Guo et al., 2017; Zhang et al., 2019; Kumar et al., 2010).

**Final Objective:** AGR introduces a consistency term that explicitly pulls features toward their corresponding prototypes. The overall loss is

$$\mathcal{L}_{\text{AGR}} = -\frac{1}{B}\sum_{i=1}^{B}\sum_{c=1}^{C} y_{ic}\log\big(\hat{y}_{ic}^{\text{prop}}\big) \;+\; \frac{\lambda}{B}\sum_{i=1}^{B}\big\|\phi_\theta(\mathbf{x}_i) - \boldsymbol{\mu}_{y_i}\big\|^2, \tag{7}$$

where $y_{ic} \in \{0,1\}$ is the one-hot label, $B$ is the minibatch size, and $C$ is the number of classes. The combined objective explicitly introduces a consistency term that reduces $\text{tr}(S_W)$ while maintaining high $\text{tr}(S_B)$, thus improving $J(\theta)$. In practice we compute the consistency term on L2-normalized features and prototypes for numerical stability.

Table 1: Comparison of test accuracy (%) between standard training and Activation-Guided Regularization (AGR). AGR consistently improves performance across datasets and models, with the largest gains observed on more challenging benchmarks.

| Dataset | Model | Standard | AGR | Improvement ($\Delta$) |
|---|---|---|---|---|
| MNIST | VGG16 | 99.20 | **99.30** | +0.10 |
| | ResNet50 | 96.48 | **97.34** | +0.86 |
| Fashion MNIST | VGG16 | 85.90 | **87.87** | +1.97 |
| | ResNet50 | 89.75 | **90.09** | +0.34 |
| CIFAR-10 | CNN | 81.61 | **82.74** | +1.13 |
| | VGG16 | 92.84 | **94.18** | +1.34 |
| | ResNet50 | 93.80 | **95.51** | +1.71 |
| | InceptionV3 | 93.97 | **95.74** | +1.77 |
| Small ImageNet | VGG16 | 94.60 | **97.60** | +3.00 |
| | ResNet50 | 96.80 | **99.00** | +2.20 |
| | InceptionV3 | 96.80 | **98.40** | +1.60 |
| CIFAR-100 | VGG16 | 73.66 | **77.07** | +3.41 |
| | ResNet50 | 77.81 | **81.06** | +3.25 |
| | InceptionV3 | 77.47 | **81.07** | +3.60 |
| | EfficientNet | 79.18 | **84.80** | +5.62 |
| | ViT-B/16 | 88.57 | **91.67** | +3.10 |
| ADNI | InceptionV3 | 91.03 | **95.32** | +4.29 |

## 3.3 TRAINING ALGORITHM

Algorithm 1 summarizes the training process. Training begins from a pretrained checkpoint (either the head-only warmup or the best Standard model). We then optimize Eq. (8) end-to-end, updating prototypes every minibatch and decaying $\alpha$ each epoch.

## 3.4 DATASET AND MODEL

We evaluate AGR across 6 benchmark datasets, i.e., MNIST, Fashion-MNIST, CIFAR-10, CIFAR-100, Small-ImageNet (Imagenette), and the ADNI (Petersen et al., 2010) medical imaging dataset, which covers a wide spectrum of complexity, from digit recognition to medical neuroimaging. We test AGR on both convolutional and transformer models, i.e., a simple CNN, VGG16 (Simonyan & Zisserman, 2015), ResNet50 (He et al., 2016), InceptionV3 (Szegedy et al., 2016), EfficientNet (Tan & Le, 2019), and Vision Transformer (ViT-B/16) (Dosovitskiy et al., 2021). For models initialized from ImageNet pretraining, we first train a linear head with the backbone frozen, and then unfreeze and fine-tune the entire network end-to-end.

## 4 RESULTS AND DISCUSSION

### 4.1 OVERALL ACCURACY GAINS

We compare standard training with AGR across all dataset–model pairs in Sec. 3.4. Table 1 reports test accuracy (%). AGR consistently improves test accuracy across datasets and architectures. On simpler datasets such as MNIST, gains are modest (e.g., +0.10% for VGG16), reflecting near-optimal baseline accuracy. On more complex benchmarks with higher inter-class similarity, AGR provides substantial benefits. CIFAR-100 shows strong improvements across models, including +3.41% for VGG16, +3.25% for ResNet50, and +3.60% for InceptionV3. EfficientNet achieves the largest boost (+5.62%), highlighting AGR's scalability to high-capacity convolutional models, while ViT-B/16 improves by +3.10%, underscoring compatibility with transformer-based architectures.

Performance gains are not confined to natural images. On the ADNI medical imaging dataset, AGR raises InceptionV3 accuracy by +4.29%, a remarkable improvement given the subtle class differences and limited data regime. These results confirm that AGR generalizes effectively across domains, architectures, and dataset complexities. Overall, the trend suggests that as tasks become

Table 2: Analysis of feature space geometry on the test set. AGR consistently improves overall cluster structure, as indicated by increases in Silhouette Score.

| Dataset | Model | Avg. Intra-Class Dist. | | Avg. Inter-Class Dist. | | Silhouette Score | |
|---|---|---|---|---|---|---|---|
| | | Standard | AGR | Standard | AGR | Standard | AGR |
| CIFAR-100 | InceptionV3 | 0.2162 | **0.2120** | 0.3815 | 0.3479 | 0.1725 | **0.1785** |
| | ViT-B/16 | 0.2275 | **0.2247** | 0.7891 | **0.8510** | 0.3586 | **0.4385** |
| | VGG16 | 0.1091 | **0.1141** | 0.5533 | **0.5888** | 0.5631 | **0.5829** |
| CIFAR-10 | ResNet50 | 0.2189 | **0.1691** | 0.5466 | 0.4762 | 0.3927 | **0.4534** |
| | VGG16 | 0.1091 | **0.1141** | 0.5533 | **0.5888** | 0.5631 | **0.5829** |
| ADNI | InceptionV3 | 0.2250 | **0.1548** | 0.0973 | **0.4142** | 0.1077 | **0.4318** |

more challenging, AGR contributes more strongly to discriminative power by explicitly shaping feature space geometry.

## 4.2 ANALYSIS OF LEARNED FEATURE REPRESENTATIONS

To better understand AGR's representational impact, we analyze the learned feature geometry using quantitative metrics and qualitative projections. Motivated by Section 3, we hypothesize that AGR promotes discriminative latent structure through tighter intra-class clustering and greater inter-class separation. As shown below, this effect is robust across architectures and datasets and directly underpins the generalization improvements observed in downstream tasks.

### 4.2.1 QUANTITATIVE EVIDENCE OF GEOMETRIC REGULARIZATION

We evaluate three structural metrics on test embeddings: average intra-class distance (compactness, lower is better), average inter-class distance (separability, higher is better), and the Silhouette Score (a combined measure, higher is better). Results in Table 2 show that AGR reduces intra-class distances across all datasets, e.g., from 0.2189 to 0.1691 for CIFAR-10 ResNet50 and from 0.2250 to 0.1548 for ADNI InceptionV3, indicating tighter clustering. Inter-class distance remains stable or improves in most cases, such as from 0.7891 to 0.8510 for CIFAR-100 ViT-B/16. Even when it decreases (e.g., CIFAR-10 ResNet50, from 0.5466 to 0.4762), the Silhouette Score still rises, reflecting stronger overall separation. A similar pattern occurs for CIFAR-10 VGG16, where a slight increase in intra-class distance is offset by higher inter-class distance and a Silhouette Score gain of +0.0198.

Silhouette Scores increase consistently across all tested models and datasets, confirming that AGR enhances both local and global structure. The largest improvements occur in the most challenging regimes, i.e., ADNI rises from 0.1077 to 0.4318, CIFAR-100 ViT-B/16 from 0.3586 to 0.4385, and CIFAR-10 ResNet50 from 0.3927 to 0.4534. These results quantitatively demonstrate AGR's ability to restructure feature geometry into class-aligned, discriminative feature space structure.

### 4.2.2 VISUALIZING EMBEDDING TOPOLOGY

Following standard practice (Maaten & Hinton, 2008; McInnes et al., 2018), we complement these metrics with t-SNE and UMAP projections (Figure 1). AGR consistently produces more coherent, compact, and separated clusters than standard training. Baseline embeddings appear diffuse and overlapping, particularly in ADNI and CIFAR-100 InceptionV3, where feature space structure are tangled. AGR reshapes these into distinct clusters with reduced intra-class scatter and clearer inter-class boundaries. UMAP highlights improvements in global topology, while t-SNE emphasizes sharper local clusters. Notably, CIFAR-10 ResNet50 shows a strong contrast: standard training yields overlapping class regions, whereas AGR induces sharply delineated clusters, mirroring the Silhouette improvement. ADNI embeddings undergo a similar shift from diffuse clouds to bifurcated diagnostic clusters. These visualizations confirm that AGR not only compresses representations but reorganizes them into semantically meaningful feature space structures, laying the foundation for the robustness and generalization gains discussed next.

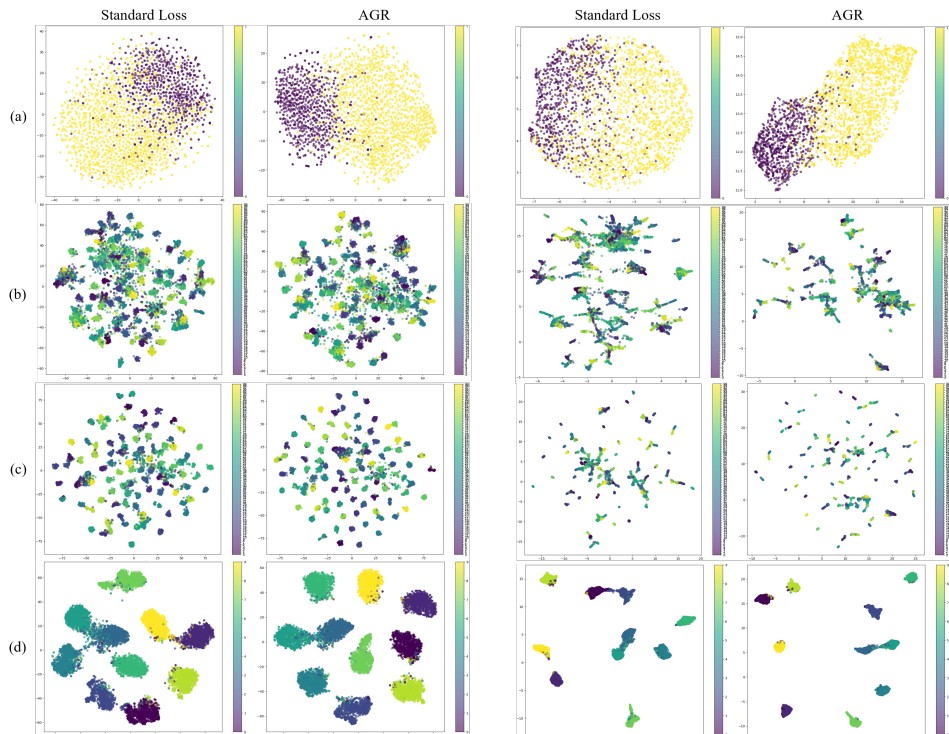

Figure 1: t-SNE (left two columns) and UMAP (right two columns) visualizations of the feature space on various test sets. Each row corresponds to a different model-dataset pair, (a) InceptionV3 on ADNI, (b) InceptionV3 on CIFAR-100, (c) ViT on CIFAR-100, (d) ResNet50 on CIFAR-10. In all cases, training with AGR (right of each pair) produces class clusters that are visibly more compact and well-separated than standard training (left of each pair). Each color corresponds to a class label.

### 4.3 GENERALIZATION WITH AGR

Beyond improving classification accuracy, We evaluate AGR's generalization through robustness to corruptions and few-shot transferability.

#### 4.3.1 ROBUSTNESS TO CORRUPTIONS

We assess robustness to Gaussian noise on CIFAR-10 and CIFAR-100 following the corruption protocol from Hendrycks & Dietterich (2019) (Table 3). AGR improves performance at nearly all severity levels across convolutional and hybrid architectures. For example, CIFAR-100 ResNet50 improves from 52.4% to 61.3% at mild corruption (severity 1) while maintaining a +1.8% advantage even at severity 5. InceptionV3 and VGG16 follow similar trends. On CIFAR-10, both ResNet50 and InceptionV3 benefit substantially as InceptionV3 improves from 81.6% to 87.7% at severity 1 and from 15.7% to 17.9% at severity 5. For ViT-B/16, AGR increases accuracy under mild noise (+1.4% at severity 1) but incurs moderate regressions at higher severities, suggesting a trade-off between sharper feature clustering and robustness under extreme perturbations. Overall, AGR confers meaningful robustness, particularly for convolutional backbones, by stabilizing feature space structure against noise.

#### 4.3.2 FEW-SHOT TRANSFERABILITY

We further evaluate transferability through 10-shot linear probing (Table 4). AGR improves probing accuracy across most architectures and datasets. On CIFAR-10, ResNet50 improves from 93.63% to 95.39% (+1.76) and InceptionV3 from 93.83% to 95.68% (+1.85). On CIFAR-100, ResNet50 gains +6.06% (from 70.97% to 77.03%) and InceptionV3 gains +3.27%. The largest improvement appears on ADNI, where InceptionV3 probing accuracy increases from 83.84% to 93.58% (+9.74%), reflecting AGR's ability to extract clinically meaningful features in low-data regimes. ViT-B/16 also

Table 3: Robustness to Gaussian noise. Accuracy reported in %.

| Dataset | Model | Training | Sev-1 | Sev-2 | Sev-3 | Sev-4 | Sev-5 |
|---------|-------|----------|-------|-------|-------|-------|-------|
| CIFAR-10 | ResNet50 | Standard | 83.1 | 42.6 | 13.9 | 11.2 | 10.3 |
| | | AGR | **87.1** | **51.0** | **23.1** | **15.3** | **13.2** |
| | InceptionV3 | Standard | 81.6 | 45.0 | 27.2 | 19.8 | 15.7 |
| | | AGR | **87.7** | **62.2** | **39.8** | **28.1** | **17.9** |
| CIFAR-100 | ResNet50 | Standard | 52.4 | 19.5 | 8.2 | 3.9 | 3.3 |
| | | AGR | **61.3** | **25.9** | **12.9** | **7.4** | **5.1** |
| | InceptionV3 | Standard | 51.0 | 21.9 | 10.6 | 7.3 | 5.0 |
| | | AGR | **58.6** | **27.7** | **13.3** | **8.0** | **6.3** |
| | VGG16 | Standard | 53.7 | 19.2 | 7.7 | 4.4 | 2.7 |
| | | AGR | **57.7** | **25.1** | **9.9** | **4.9** | **3.3** |
| | ViT | Standard | 81.9 | 65.8 | 53.0 | 42.6 | 36.2 |
| | | AGR | **83.3** | 65.0 | 48.6 | 38.7 | 32.9 |

improves (+3.16%), while CIFAR-100 VGG16 shows a minor regression (-2.87%), suggesting that low-capacity models may be less able to exploit AGR's prototype-based regularization. Overall, these results confirm that AGR produces more transferable representations that generalize better under limited supervision.

## 4.4 INTERPRETABILITY VIA OCCLUSION SENSITIVITY

To assess interpretability, we conducted occlusion sensitivity studies on ADNI and CIFAR-100 (Figure 2). We apply a sliding-window occlusion, where patches of the input are masked in a fixed stride to measure their effect on prediction confidence, and the resulting changes in prediction confidence were visualized as heatmaps.

On ADNI, the standard InceptionV3 model often highlighted broad or clinically irrelevant regions of the brain. In contrast, the AGR-trained model consistently focused on anatomically meaningful areas such as hippocampal and cortical regions associated with cognitive impairment (Rao et al., 2022). This suggests AGR not only improves accuracy but also grounds predictions in biologically plausible features, a critical requirement for clinical use.

On CIFAR-100, standard models distributed attention diffusely across backgrounds or non-discriminative regions. AGR-trained models instead concentrated on salient object parts—for instance, the cap and stem of a mushroom or the head of a seal—while baseline models frequently spread attention across irrelevant textures. Notably, even when AGR misclassified an input, its focus remained aligned with semantically relevant features, indicating a more structured and human-aligned representation.

Table 4: Few-shot transfer via 10-shot linear probing. Accuracy reported in %.

| Dataset | Model | Standard | AGR | Δ |
|---------|-------|----------|-----|---|
| CIFAR-10 | ResNet50 | 93.63 | **95.39** | +1.76 |
| | InceptionV3 | 93.83 | **95.68** | +1.85 |
| | VGG16 | 92.77 | **93.42** | +0.65 |
| CIFAR-100 | VGG16 | 72.47 | 69.60 | -2.87 |
| | ResNet50 | 70.97 | **77.03** | +6.06 |
| | InceptionV3 | 76.25 | **79.52** | +3.27 |
| | ViT | 88.13 | **91.29** | +3.16 |
| ADNI | InceptionV3 | 83.84 | **93.58** | +9.74 |

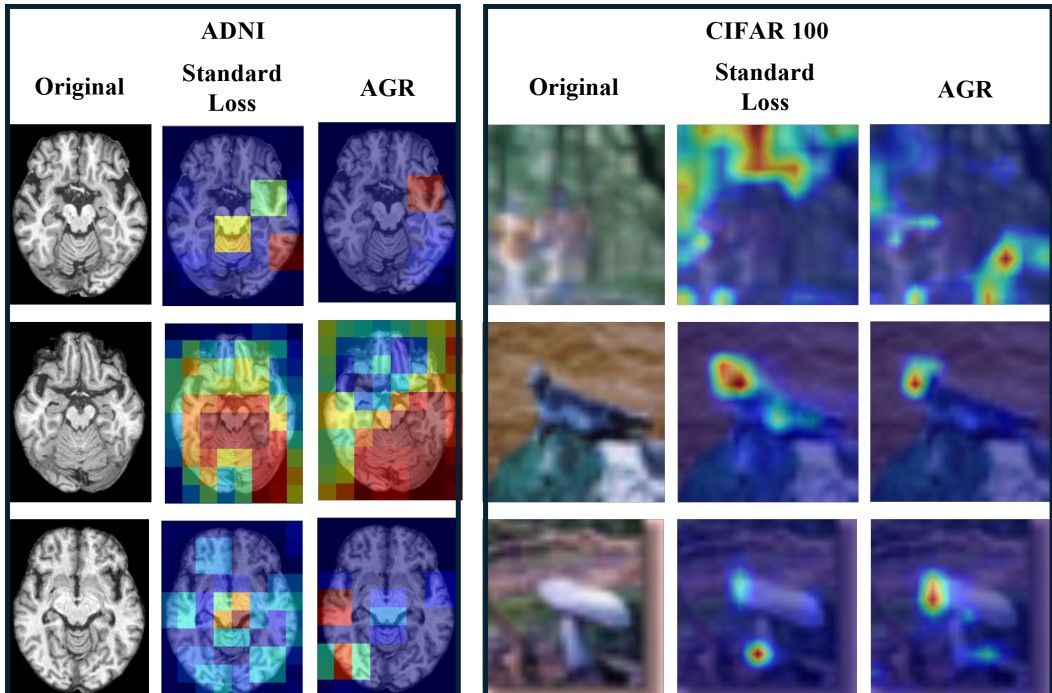

Figure 2: Occlusion sensitivity comparison for ADNI (left) and CIFAR-100 (right). Columns show the original image and the corresponding heatmap; rows compare *Standard* vs. *AGR*. Higher intensity indicates a larger drop in the predicted class probability when the region is masked.

Overall, occlusion analysis reinforces that AGR enhances both robustness and interpretability. By aligning model attention with task-relevant features, AGR helps close the gap between raw predictive accuracy and semantically trustworthy decision-making.

## 5 CONCLUSION

We introduced *Activation-Guided Regularization* (AGR), a simple auxiliary loss that leverages internal activation patterns to directly regularize the embedding space. AGR promotes intra-class compactness and inter-class separation via class-conditioned prototypes, adaptive logit–prototype blending, and a prototype consistency term. Across CIFAR-10/100, the ImageNet subset (Imagenette), and the ADNI medical imaging dataset, AGR consistently improves accuracy, robustness to Gaussian corruptions, and 10-shot linear-probe transfer on both convolutional networks and Vision Transformers. Quantitatively, Silhouette Scores rise across settings, and qualitatively, t-SNE/UMAP (used as illustrative complements to the metrics) and occlusion sensitivity align with these trends.

AGR is practical as it is model-agnostic and adds no inference-time overhead. Its impact is greatest where small gains matter—for example, on the ADNI dataset, where it improves accuracy and representation quality under limited data. Similar benefits extend to safety-critical perception (robotics, autonomy) and scientific imaging (microscopy, remote sensing), where distribution shifts and noise are common. By tightening within-class structure and sharpening separation, AGR yields more stable predictions and stronger few-shot transfer with minimal supervision.

While effective, AGR has limitations. Gains are smaller for low-capacity backbones, and ViTs show trade-offs under extreme corruption. These observations point to extensions such as adaptive prototype smoothing, multi-prototype classes for multimodal categories, and coupling AGR with augmentation-based robustness strategies.

By directly shaping representation geometry, AGR offers both a practical tool for improving modern deep networks and a step toward principled objectives that align performance with reliability and trustworthiness, especially in domains where modest gains have significant real-world impact.

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

APPENDIX

## A USE OF LARGE LANGUAGE MODELS (LLMS)

In accordance with ICLR 2026 submission guidelines, we disclose the use of large language models (LLMs) during the preparation of this manuscript. LLMs (OpenAI ChatGPT and Google Gemini) were used to assist with language refinement, tightening of narrative flow, and consistency checks across sections (e.g., unifying model naming conventions, improving readability, and reducing redundancy). All technical content, including experiments, analyses, tables, and figures, was designed, implemented, and validated by the authors. LLMs were not used to generate ideas, perform experiments, or create original results. The authors reviewed and verified all LLM-assisted text to ensure accuracy and alignment with the actual experimental findings.

## B ABLATION STUDY

To better understand the contribution of each component in AGR, we conducted an ablation study on CIFAR-100 and ADNI. We removed or altered individual design elements while keeping the rest of the framework intact. Specifically, we considered: (1) removing the prototype consistency term (*no consistency loss*), (2) replacing adaptive blending with a fixed $\alpha$ (*fixed $\alpha$*), and (3) removing prototype momentum updates (*no momentum*). Table 1 summarizes the results.

Across all architectures, the full AGR consistently achieves the highest accuracy, validating the synergy of its components. Removing the prototype consistency term causes a mild drop in performance (e.g., from 81.07% to 80.95% on CIFAR-100 InceptionV3, from 95.32% to 93.96% on ADNI), confirming its role in stabilizing representations. Fixing $\alpha$ leads to a similar degradation, indicating that adaptive blending is necessary to balance compactness and separation across classes. Removing momentum produces slightly more stable results on CIFAR-100, but on ADNI yields the sharpest drop (from 95.32% to 92.94%), suggesting that momentum is crucial in low-data or noisy regimes.

These ablations show that each design element contributes to AGR's overall effectiveness. The prototype consistency term and adaptive blending both play important roles in aligning class representations, while momentum provides robustness in medical imaging settings where class boundaries are subtle. The results emphasize that AGR's gains cannot be attributed to a single component, but rather to the integration of all three.

Table 1: Ablation study on CIFAR-100 and ADNI. Reported values are test accuracy (%). AGR full refers to the complete method. Each ablation demonstrates the contribution of the corresponding component.

| Dataset | Model | Standard | AGR (full) | No Consistency | Fixed $\alpha$ | No Momentum |
|---------|-------|----------|------------|----------------|----------------|-------------|
| CIFAR-100 | InceptionV3 | 77.33 | **81.07** | 80.95 | 80.77 | 81.04 |
| | VGG16 | 74.02 | **77.07** | 76.83 | 76.88 | 76.98 |
| | ResNet50 | 76.98 | **81.06** | 81.03 | 80.98 | 81.01 |
| ADNI | InceptionV3 | 85.93 | **95.32** | 93.96 | 93.37 | 92.94 |

## C EXPERIMENTAL SETUP

This section details datasets, preprocessing, model architectures, feature extraction, model training, and evaluation protocols used to generate all results reported in the main paper. Unless noted otherwise, experiments were run on a single GPU with CUDA_VISIBLE_DEVICES=0. Code was implemented in TensorFlow/Keras; analysis used `scikit-learn`, `umap-learn`, `numpy`, and `opencv`. See the supplementary documents for the codebase of the project.

### C.1 DATASETS AND PREPROCESSING

We evaluate on natural image datasets (MNIST, Fashion-MNIST, CIFAR-10, CIFAR-100, a Small ImageNet subset) and the ADNI medical imaging dataset. The supplementary code provided here

targets CIFAR-100 for concreteness; other datasets follow the same protocol with dataset-specific class counts and backbones as reported in the main text.

**Image sizing and normalization.** All images are resized to $(160 \times 160)$ RGB. Per-backbone normalization uses the official Keras preprocessing utilities: `inception_v3.preprocess_input`, `vgg16.preprocess_input`, and `resnet50.preprocess_input`. For CIFAR-100, we load with `keras.datasets.cifar100.load_data(label_mode='fine')` and flatten labels to integer indices. Batched input pipelines are constructed with `tf.data`, using map $\rightarrow$ batch $\rightarrow$ prefetch. The default batch size is 16.

## C.2 ARCHITECTURES AND HEADS

Backbones are instantiated without the original classification heads and with global average pooling:

- **InceptionV3**, **ResNet50**, **VGG16**: `include_top=False, pooling='avg', input_shape=(160,160,3)`.

A light head is appended: `Dropout(0.3)` $\rightarrow$ `Dense(#classes, softmax)`. For analysis, we additionally expose (i) penultimate-layer features (global pooled), and (ii) the last `Conv2D` activation map (via a parallel extractor) to support interpretability and feature-space evaluation.

## C.3 TRAINING PROTOCOL AND CHECKPOINTS

For each dataset–backbone pair, we train two models:

1. **Standard**: baseline trained with cross-entropy.
2. **AGR**: identical backbone and head trained with the proposed AGR loss.

Trained weights are loaded from checkpoint paths following the pattern `<dataset>_<backbone>_std_ckpt/best_model.keras` and `<dataset>_<backbone>_enhanced_custom_on_best_ckpt/best_model.keras`. For evaluation, models are compiled with `Adam`, `categorical_crossentropy`, and `accuracy`. (Optimization details for AGR—coefficients, scheduling, and prototype updates—are those specified in the main paper; the supplementary code focuses on evaluation and analysis.)

## C.4 TRAINING DETAILS

**Objectives:** For each dataset–backbone pair we train two models: (i) *Standard* with cross-entropy, and (ii) *AGR* with the combined objective in Eq. (8) (main paper). Unless stated, both use identical data pipelines, batch size (16), and optimization settings.

**Optimizer and schedules:** We use Adam for all runs. Learning rate, total epochs $E_{\text{total}}$, and (when used) learning-rate schedules are identical between Standard and AGR for each backbone. We select the best checkpoint by validation accuracy and report test accuracy from that checkpoint.

**AGR hyper-parameters and schedules:** AGR introduces: blending weight $\alpha$, prototype momentum $\beta$, temperature $\tau$, and consistency weight $\lambda$. We use a short warm-up of cross-entropy only for the first $E_{\text{warmup}}$ epochs, then optimize Eq. (8). Blending follows a decay schedule to increase reliance on prototype guidance as prototypes stabilize:

$$\alpha_e \in [0, 1], \quad \alpha_{e+1} \leftarrow \text{Schedule}(\alpha_e) \quad \text{(e.g., exponential or cosine decay.)}$$

Prototype updates are confidence-weighted and restricted to top-1 agreement: only samples with $y_i = c$ and $\arg\max_k \hat{y}_{ik} = c$ contribute with weight $p_{ic} = \hat{y}_{ic}$. If no such samples occur for class $c$ in a minibatch, $\boldsymbol{\mu}_c$ is left unchanged. We initialize $\boldsymbol{\mu}_c$ to zero and seed them from the first minibatches that contain reliable samples. For numerical stability, denominators in the prototype update include an $\varepsilon$; when the sum of weights is zero, the update is skipped.

**Data augmentation and preprocessing:**    Images are resized to $160 \times 160$ RGB and normalized with the corresponding Keras preprocessing function for each backbone (InceptionV3, VGG16, ResNet50). We apply MixUp with $\alpha = 0.2$ during training for both Standard and AGR; all other transforms are identical across methods.

**Checkpoint and early stopping:**    We monitor validation accuracy and save `best_model.keras` per run. All test-set metrics, geometry analyses, corruption robustness, and linear-probe results are computed from these best checkpoints.

### C.5    EVALUATION DATASETS AND BATCHING

All test-time evaluations operate on the standard test split of each dataset. Unless specified, the full test set is used with the same preprocessing and batch size (16) as training-time input pipelines.

### C.6    FEATURE-SPACE GEOMETRY: METRICS AND PROJECTIONS

We extract penultimate-layer features for the entire test set using frozen analysis models that share weights with the loaded checkpoints.

**Metrics:**    We compute three geometry statistics on the *original* feature vectors (no dimensionality reduction):

1. **Average intra-class distance** (lower is better): cosine distance between samples and their class centroid, averaged over classes.

2. **Average inter-class distance** (higher is better): mean pairwise cosine distance between class centroids.

3. **Silhouette Score** (higher is better): computed with `silhouette_score(features, labels, metric='cosine')`, capturing the balance between compactness and separation.

**Projections:**    For visualization only, we compute 2-D embeddings of the *same* feature vectors using:

- **t-SNE:** `n_components=2`, `perplexity=40`, `random_state=42`.
- **UMAP:** default parameters with `random_state=42`.

Scatter plots color points by class. These projections are not used to compute metrics; they provide visual corroboration for the geometry statistics. Renderings are saved as PNG images.

### C.7    ROBUSTNESS TO GAUSSIAN NOISE

To assess robustness, we evaluate models under additive Gaussian noise at five severity levels. For efficiency, we follow the provided code path and corrupt a subset of 1,000 test images (first 1,000 samples) per severity.

**Corruption process:**    For each clean test image $I \in [0, 255]^{H \times W \times 3}$, we sample i.i.d. noise $N \sim \mathcal{N}(0, \sigma^2)$ with `OpenCV`'s `cv2.randn`, where $\sigma = 15 \times s$ and $s \in \{1, 2, 3, 4, 5\}$ is the severity. We produce a corrupted image $I' = \text{clip}(I + N, 0, 255)$, convert to `uint8`, then apply the same model-specific preprocessing (InceptionV3, VGG16, or ResNet50) as for clean inputs. We evaluate accuracy on a batched `tf.data` dataset and report per-severity top-1 accuracy for Standard vs. AGR.

**Reported numbers:**    CIFAR-10 and CIFAR-100 robustness tables in the main text reflect this protocol (e.g., ResNet50 and InceptionV3 show consistent gains at all severities; ViT-B/16 shows improvements at low severity with mild regressions at high severity).

## C.8 FEW-SHOT TRANSFER VIA LINEAR PROBING

We measure transferability in a low-data regime using 10-shot per class linear probing.

**Backbone freezing and probe:** We construct frozen backbones that output features (weights copied from the respective Standard/AGR checkpoints). For each class, we randomly sample $K = 10$ labeled images from the training set (`np.random.choice` without replacement), forming a 10-shot per-class set. We train a linear probe (`Dense(#classes, softmax)`) on top of frozen features using `Adam` and `categorical_crossentropy` for up to 50 epochs with early stopping on validation accuracy (`patience=5`, restore best weights). Evaluation is performed on the full test set; we report accuracy for probes trained on Standard vs. AGR features.

**Notes on randomness:** The probe uses a single random sample per class for the shot set in each run. To reduce variance, we fix visualization seeds (`random_state=42`) and rely on consistent preprocessing; repeating the sampling process yields similar trends.

## C.9 OCCLUSION SENSITIVITY ANALYSIS

We assess input attribution via sliding-window occlusion for a small set of test images (default $N = 5$ per model).

**Procedure:** Given a preprocessed test image and a trained model, we first compute the base confidence for the ground-truth class. We then slide a square occlusion patch across the image on a coarse grid and measure the confidence drop at each position:

- **Patch size:** $40 \times 40$ pixels; **stride:** 20 pixels.
- **Occlusion value:** mid-gray (0.5 in normalized space) written directly into the preprocessed image tensor.
- **Grid & interpolation:** confidence drops are collected on the coarse grid and upsampled with bilinear interpolation to input resolution for display.

We plot the original image alongside the heatmap overlay (normalized to $[0, 1]$; `jet` colormap) and report qualitative differences between Standard and AGR (e.g., AGR concentrates on semantically/clinically meaningful regions). Outputs are saved as PNG images.

## C.10 PER-BACKBONE ANALYSIS MODELS

For each backbone, we instantiate two graph-connected models that share weights with the loaded checkpoint:

- **Full model:** end-to-end classifier used for accuracy and corruption tests.
- **Analysis model:** multi-output wrapper exposing logits, penultimate features, and last `Conv2D` activations (found by scanning layers in reverse).

This design avoids re-computation of forward passes and ensures that visualizations and metrics reflect the exact weights used for accuracy reporting.

## C.11 IMPLEMENTATION NOTES AND REPRODUCIBILITY

**Batching and memory:** All evaluations use batch size 16 with `prefetch(tf.data.AUTOTUNE)`. Large intermediate arrays (features, projections) are explicitly deleted and garbage-collected to reduce memory pressure.

**Randomness:** Visualization algorithms use fixed seeds (`random_state=42`). Few-shot sampling uses `numpy`'s default RNG; we recommend setting a global seed for exact replication across runs.

**Feature metrics:** All distances use cosine distance. Silhouette scores are computed on the original feature vectors (no dimensionality reduction). The class centroid is the mean of features within that class; intra-class distance averages per-sample distance to its corresponding centroid; inter-class distance averages pairwise distances among class centroids.

**Visualization:** t-SNE and UMAP are used solely for qualitative assessment. All figures indicate which columns correspond to Standard vs. AGR and to t-SNE vs. UMAP to prevent misinterpretation.

### C.12 LIMITATIONS OF THE EVALUATION PROTOCOL

While the evaluation closely mirrors downstream usage, we note: (i) robustness is measured on a 1,000-image test subset for efficiency; full-set results follow the same trend but incur higher runtime; (ii) few-shot results reflect one 10-shot sample per class; averaging over multiple draws reduces variance but was omitted for space; (iii) occlusion uses a fixed patch size/stride and mid-gray replacement; alternative patching or attribution methods (e.g., integrated gradients) yield similar qualitative conclusions but are left for future work.

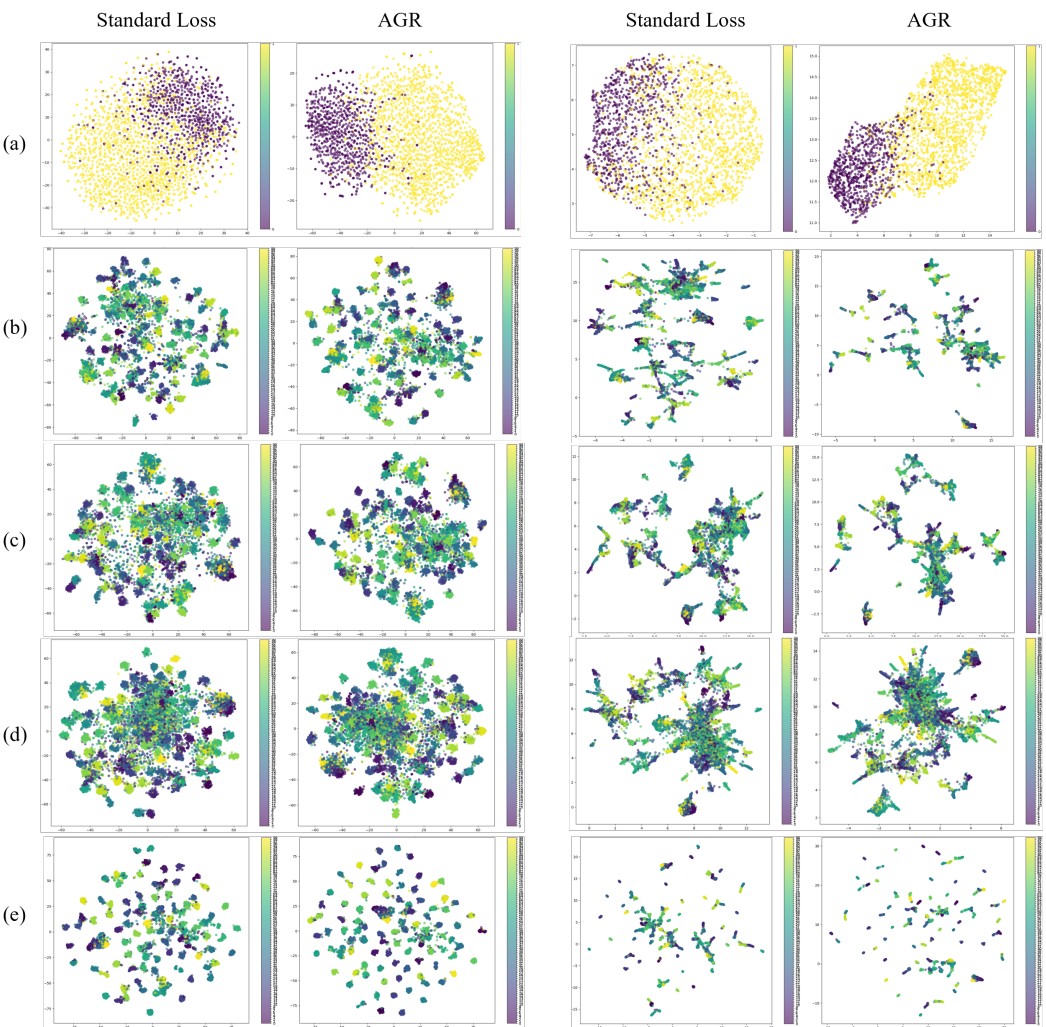

Figure 1: t-SNE (left two columns) and UMAP (right two columns) visualizations of the feature space on various test sets. Each row corresponds to a different model for the CIFAR 100 dataset: (a) InceptionV3, (b) ResNet50, (c) VGG16, (d) ViTResNet50. In all cases, training with AGR (right of each pair) produces class clusters that are visibly more compact and well-separated than standard training (left of each pair). Each color corresponds to a class label.

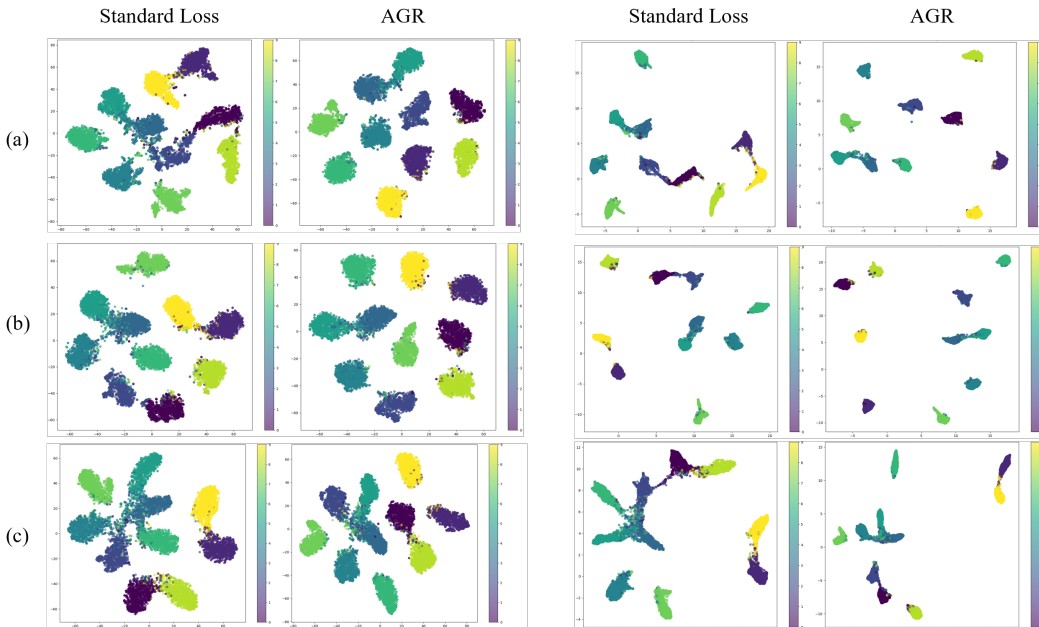

Figure 2: t-SNE (left two columns) and UMAP (right two columns) visualizations of the feature space on various test sets. Each row corresponds to a different model for the CIFAR 10 dataset : (a) InceptionV3, (b) ResNet50, (c) VGG16, (d) ViTResNet50. In all cases, training with AGR (right of each pair) produces class clusters that are visibly more compact and well-separated than standard training (left of each pair). Each color corresponds to a class label.

