# OpenReview forum: "Activation-Guided Regularization: Improving Deep Classifiers using Feature-Space Regularization with Dynamic Prototypes"
_ICLR.cc/2026/Conference — ICLR 2026 Conference Withdrawn Submission_

### Official Review · Reviewer_onMg · 2025-10-27

**Soundness:** 2
**Presentation:** 3
**Contribution:** 2
**Rating:** 2
**Confidence:** 4

**Summary:**

This paper proposed AGR, a loss function same to CELoss, which can significantly improve the intra-class and inter-class distributions.

**Strengths:**

1.In a considerable number of experiments on multiple datasets, AGR has shown significant improvements.
2.The author provides a certain degree of visualization.

**Weaknesses:**

1.The author claims that AGR" can always be improved ". However, in some experiments, AGR played a significant negative role. This statement is inaccurate, 'low-capacity models' should be clearly defined and sufficient experiments provided to show readers the clear scope of application.
2.The superiority of the proposed method should come from comparison. In fact, there are many excellent works on self-supervised or regular loss (the author has also listed them in related works). Although the implementation ideas may not be consistent, their inclusion can still enhance persuasiveness.
3.The lack of key information such as GPU model will significantly affect reproducibility.
4.In terms of writing, the author should provide clear contribution points and more illustrations to help readers understand quickly. This is allowed in terms of the length of the paper.
5.In section 3.2, the paragraph abstract It sometimes ends with ". "and sometimes with" : ".In page 19, caption of Fig.2, line 3 and In page 18, caption of Fig.1, line 3 what is ‘ViTResNet50’?  In Algorithm 1, y_b should also be in required.
6.Some unclear edits. Missing line numbers, incorrect references to some formulas and charts, manifested as inability to link.

**Questions:**

1.In Algorithm 1, it is clearly stated that CELoss remains a part of the computational steps. May I ask if AGR is a weighted combination of the regularization term and CE, or an equivalent term of CE?
2.Can the author re-explain the necessity of the word "activation" in the naming? Clarifying this process is very helpful for understanding.
3.Can the author demonstrate the performance on larger datasets (e.g., ImageNet1K)?
If the author can clearly explain the reasons for the current experimental design, I will consider improving my score.

---

### Official Review · Reviewer_mXpr · 2025-10-30

**Soundness:** 3
**Presentation:** 3
**Contribution:** 1
**Rating:** 2
**Confidence:** 4

**Summary:**

This work introduces Activation Guided Regularization (AGR). AGR combines standard cross-entropy training with a secondary ojbective that encourages a sample's features to be similar to a prototype computed on-the-fly. This work finds that this improves performance over several architechtures and model classes.

**Strengths:**

* The paper is clearly written.
* The idea is both reasonable and straighforward to implement, increasing practicality.
* The results consistently improve over standard cross-entropy on the datasets tested.

**Weaknesses:**

* There is a substantial lack of of comparison to baselines. Supervised Contrastive Learning also results in much tighter clusters, for example. Even though it comes at the cost of using a larger batch size with pair-wise comparisons, it is worth comparing to. Similarly, label smoothing has been show to have a very similar effect (tightening class clusters) in [2]. I believe this is a major issue in being able to understand how well this works.

* In Section 3.3, the authors state that training begins from a pre-trained checkpoint (the best standard model).

* The datasets are quite small for assessment and perhaps close to saturated; many accuracies are over 90%.




[1] Khosla, Prannay, et al. "Supervised contrastive learning." Advances in neural information processing systems 33 (2020): 18661-18673.
[2] Müller, Rafael, Simon Kornblith, and Geoffrey E. Hinton. "When does label smoothing help?." Advances in neural information processing systems 32 (2019).

**Questions:**

* Could the authors discuss competing methods such as supervised contrastive learning and label smoothing, and maybe show some results? [1].[2]
* How would performance change for standard cross entropy under similar continued training as AGR?
* Could the authors try on a harder dataset, maybe full ImageNet?

[1] Khosla, Prannay, et al. "Supervised contrastive learning." Advances in neural information processing systems 33 (2020): 18661-18673.
[2] Müller, Rafael, Simon Kornblith, and Geoffrey E. Hinton. "When does label smoothing help?." Advances in neural information processing systems 32 (2019).

---

### Official Review · Reviewer_bWxR · 2025-10-30

**Soundness:** 3
**Presentation:** 4
**Contribution:** 3
**Rating:** 4
**Confidence:** 3

**Summary:**

The paper introduces a technique dubbed Activation-Guided Regularisation (AGR). AGR consists of two key components: (i) a regularisation term that encourages the last-layer activations to align with some class prototypes, which are defined self-consistently from the activations of correctly classified samples, and (ii) the replacement of the standard prediction with a linear combination of the network’s logits and a similarity-based prediction derived from the class prototypes. The authors demonstrate that AGR enhances classification accuracy across standard image classificatio benchmarks. They further investigate its effect on representation learning by analysing intra- and inter-class distances in the learned feature space, and by providing t-SNE and UMAP visualisations of the resulting embeddings. Additional experiments show that AGR leads to improved robustness to input corruptions, few-shot transfer performance, and interpretability of learned representations.

**Strengths:**

- The paper is clearly written and easy to follow;
- The proposed method is convincing---the intuitive explanation in Section 3.1 is particularly appreciated---and it seems straightforward to implement;
- The evaluation is comprehensive, covering standard performance on multiple vision benchmarks as well as additional desirable aspects such as robustness and interpretability.

**Weaknesses:**

1. The paper lacks references or comparisons for the reported benchmark performances. Also, I would like to know how the method compares with other regularisation schemes (e.g., SAM).
2. The evaluation does not include ImageNet, which is a widely accessible and standard benchmark. Given that many vision models are pre-trained on ImageNet before being adapted to other datasets, such a comparison would be valuable.
3. The visualisation of the learned embeddings is unconvincing: the left and right columns in both groups look very similar, making it hard to see any difference. A quantitative metric could provide a more reliable evaluation.
4. The setup of the few-shot transferability experiment is unclear (possibly not explained at all), and the occlusion sensitivity analysis lacks a quantitative assessment.

**Questions:**

All my questions are directly connected to the weaknesses, and I will raise my score if they are addressed convincingly.

---

### Official Review · Reviewer_v3Ez · 2025-11-04

**Soundness:** 2
**Presentation:** 2
**Contribution:** 2
**Rating:** 2
**Confidence:** 4

**Summary:**

This paper proposes Activation-Guided Regularization (AGR). It augments standard cross-entropy (CE) training by introducing a prototype-driven similarity distribution: the model’s prediction and a distribution from temperature softmax over cosine similarities between features and class prototypes are linearly mixed with coefficient α, and CE is computed on this mixed distribution. An additional L2 consistency term pulls features toward their class prototype to improve intra-class compactness and inter-class separation (with a Fisher-style motivation). The training schedule employs a warm-up phase with CE only, followed by updates to prototypes per batch using high-confidence samples via a confidence-weighted centroid and momentum. This is then followed by the application of the mixed prediction and consistency regularizer. AGR leaves the classifier head unchanged, requires no pair mining, and can be plugged into existing pipelines as a lightweight, architecture-agnostic regularizer. Experiments on CIFAR-10/100 and ADNI report gains in accuracy, noise robustness, and 10-shot linear probing, suggesting that AGR learns more transferable representations and focuses on more semantic regions.

**Strengths:**

- Lightweight and easy to deploy: AGR is an architecture-agnostic regularizer that uses nonparametric class prototypes, requires no pair mining, keeps the classifier head unchanged, and drops into existing training pipelines with low engineering cost.

- Clear motivation and formulation: the method links improvements to intra-class compactness and inter-class separation, formalizes a mixed probability cross-entropy with a coefficient alpha, and adds an L2 consistency term that pulls features toward class prototypes in a way consistent with a Fisher-style objective.

- Strong qualitative evidence: feature space visualizations show tighter and more separable clusters, and attention or activation maps focus on semantically meaningful regions, supporting the claim that AGR improves representation geometry.

**Weaknesses:**

- Omission of the dynamic prototype update math formula and part of the implementation details

The paper relies on vague terms like confidence-weighted centroid and momentum update, but omits the actual equation. The momentum parameter ($\beta$), the exact confidence-weighting function, and the final EMA formula are all absent. This omission makes the core innovation a black box.

Meanwhile, there is a logically impossible update condition. The authors state that a prototype update is triggered only when at least $m=50$ high-confidence samples for a class are present in the mini-batch. However, the appendix explicitly defines the default batch size as 16. It is mathematically impossible for a batch of 16 samples to contain 50 samples of any kind.

- Ambiguous metric definition in the consistency loss

The definition of the core consistency loss is critically ambiguous. Equation (7) and Algorithm 1 define the loss over the raw, un-normalized feature embeddings ($\sum ||\phi_{\theta}(x_{i})-\mu_{y_{i}}||^{2}$). However, the text immediately following Eq. (7) directly contradicts this, stating: "In practice we compute the consistency term on L2-normalized features and prototypes for numerical stability". These are two fundamentally different loss functions with different geometric implications: one minimizes Euclidean distance in the embedding space, while the other optimizes for cosine similarity on a hypersphere. The paper fails to clarify which was actually used.


- Experiments

**Missing ablations.** The paper's central claim that the L2 consistency term is the primary driver of performance is not adequately supported by the experimental design. The proposed $\mathcal {L}\_{\text{AGR}}$ is a composite objective, combining: (1) a prototype-based L2 consistency term and (2) a blended cross-entropy loss $\mathcal{L}\_{CE\_{Blended}}$, which itself acts as a powerful form of prototype-guided label smoothing or knowledge distillation. However, the provided ablation study only removes the consistency term, leaving the $\mathcal{L}\_{CE\_{Blended}}$ intact3. This test resulted in only a minor performance drop on CIFAR-100 (from 81.07% to 80.95%), suggesting that the L2 term may not be the primary contributor. The authors failed to include the most critical control experiment: Standard CE + L2 Consistency Term only. Without this, it is impossible to disentangle the gains from the feature-space regularization (the paper's claimed contribution) vs. the prediction-space regularization (a significant confounding variable).

**Insufficient baseline comparisons.** The paper evaluates AGR almost exclusively against a conventional cross-entropy baseline. This comparison is insufficient to position the work or validate its novelty. For example, the L2 consistency term is a direct descendant of prototype and proxy-based metric learning losses (e.g., Center Loss or Proxy-NCA). The $\mathcal{L}\_{CE\_{Blended}}$ component directly, somehow mimics the behavior of Label Smoothing (LS) and Knowledge Distillation (KD). These relevant baselines should be contained to determine if AGR's gains are novel or if they merely recapitulate the known benefits of existing, simpler techniques.

**Missing sensitivity analysis.** The method introduces at least four new critical hyperparameters ($\alpha$, $\beta$, $\tau$, $\lambda$), yet the paper provides no sensitivity analysis, grid search, or robustness checks for any of them.

- Minor

There are many careless typos. For example, the appendix (C.4) refers to optimizing a non-existent Eq. (8).

**Questions:**

- A critical question: The prototype update mechanism's reliance on the model's own high-confidence predictions. Does this design not create a significant risk of a self-fulfilling prophecy or a confirmation bias feedback loop? For example, if the model learns a spurious correlation in its early stages, such as associating grass with the cow class, and generates high-confidence incorrect predictions based on this error, will the AGR mechanism not erroneously crystallize this spurious feature into the class prototype, thereby reinforcing the model's initial mistake rather than correcting it?

- Given the proposed method's strong conceptual resemblance to Center Loss, could the authors elaborate further on their key distinctions? I am particularly interested in a deeper comparison of how AGR's non-parametric, confidence-weighted prototype update mechanism differs from Center Loss's gradient-based learnable centers, both in terms of theoretical optimization dynamics and the empirical characteristics of the resulting prototype/center distributions during training.

- Have the authors considered framing their method and its objectives through the lens of Neural Collapse (NC) theory? This theoretical perspective seems highly relevant to the paper's goals of enforcing intra-class compactness, and while I am not requesting such an analysis, adopting this viewpoint might help in further strengthening the paper's narrative.

---

### Note · Authors · 2025-11-19

**Comment:**

On behalf of all co-authors, I respectfully request to withdraw submission 20490, “Activation-Guided Regularization: Improving Deep Classifiers using Feature-Space Regularization with Dynamic Prototypes,” from consideration for ICLR 2026.

We thank the reviewers and area chairs for their time and comments. We are planning to address several points (e.g., clarifying the loss formulation, adding stronger baselines, and re-running experiments), which is not feasible within the current review cycle.
We appreciate several specific comments, especially those regarding the L2 consistency term and the need for additional baseline comparisons (e.g., Center Loss, label smoothing, supervised contrastive learning, and SAM). We plan to examine these points in future versions of the work.

The following points are provided to clarify reviewers' confusion about the current paper:

In our implementation, class prototypes are maintained as momentum-updated running centroids over L2-normalized features of high-confidence, correctly classified samples, not arbitrary or incorrect predictions. The update is EMA-style and confidence-weighted, with a fallback estimate for classes that receive fewer reliable samples. In this context, the parameter (m = 50) denotes a threshold on the accumulated number of reliable samples for a class, rather than a requirement per minibatch (the batch size is 16), so the update condition is not mathematically impossible. In a future revision, we plan to include the explicit update equation and a concise description of the fallback mechanism.

We intended to propose a loss function that operates on top of standard cross-entropy, rather than replacing it. AGR utilizes neuron activation patterns to construct class prototypes and reweight the training signal, thereby reinforcing samples whose features align more closely with their class prototype. This design is intended to remain compatible with other regularization techniques, rather than serving as an alternative to them. The feature-space visualizations and geometry metrics were included primarily as evidence that this activation-guided loss is influencing the representation geometry in the intended way, rather than as the main contribution themselves.

One review states that we claim AGR “can always be improved” and that in some experiments AGR “played a significant negative role.” We do not make this claim anywhere in the text. In all accuracy tables we report, AGR never reduces test accuracy relative to the corresponding standard baseline; it either improves or matches it for the dataset–backbone pairs we chose. There are settings (e.g., robustness and few-shot metrics) where gains are smaller or mixed, and the narrative around those results can be made more precise, but the quoted phrasing does not appear in the manuscript.

Another comment suggests that the combined objective is missing (“non-existent Eq. (8)”). The combined objective is fully written out in the main text (currently numbered as Eq. (7)); the appendix mistakenly refers to it as Eq. (8). This is a numbering error.

These clarifications are meant to resolve issues of clarity noted by the reviewers, not to dispute their overall evaluations. We will use this feedback to significantly improve future versions of the paper.

**Withdrawal Confirmation:**

I have read and agree with the venue's withdrawal policy on behalf of myself and my co-authors.